# The Promising Role of Intestinal Organoids in the Diagnostic Work-Up of Cystic Fibrosis Screen Positive Inconclusive Diagnosis/CFTR-Related Metabolic Syndrome (CFSPID/CRMS)

**DOI:** 10.3390/ijns11030052

**Published:** 2025-07-11

**Authors:** Noelia Rodriguez Mier, Marlies Destoop, Sacha Spelier, Anabela Santo Ramalho, Jeffrey M. Beekman, François Vermeulen, Karin M. de Winter-de Groot, Marijke Proesmans

**Affiliations:** 1CF Research Lab, Woman and Child Unit, Department of Development and Regeneration, KU Leuven, 3000 Leuven, Belgiummarijke.proesmans@uzleuven.be (M.P.); 2Department of Paediatrics, Paediatric Pulmonology, University Hospital Leuven, 3000 Leuven, Belgium; 3Department of Paediatric Pulmonology, Wilhelmina Children’s Hospital, University Medical Centre Utrecht, Utrecht University, 3585 CX Utrecht, The Netherlands; m.destoop@umcutrecht.nl (M.D.); k.m.dewinter@umcutrecht.nl (K.M.d.W.-d.G.); 4Department of Regenerative Medicine Utrecht, University Medical Center, Utrecht University, 3585 CX Utrecht, The Netherlands

**Keywords:** CFSPID, patient-derived intestinal organoids, CFTR functional assays

## Abstract

Cystic Fibrosis Screen Positive Inconclusive Diagnosis/CFTR-related Metabolic Syndrome (CFSPID/CRMS) presents a significant clinical challenge due to its variable diagnostic outcomes and uncertain disease progression. Current diagnostic strategies, including sweat chloride testing and genetic analysis fall short in delivering clear guidance for clinical decision-making and risk assessment. Here, we comment on the potential of CFTR functional tests in patient-derived intestinal organoids (PDIOs) to enhance early risk stratification in CFSPID/CRMS cases. Using four hypothetical cases based on real-world data, we illustrate diverse clinical trajectories: diagnosis of cystic fibrosis (CF), reclassification as a CFTR-related disorder (CFTR-RD), non-CF designation, and persistent diagnostic uncertainty. Organoid-based assays—such as forskolin-induced swelling (FIS), steady-state lumen area (SLA) analysis, and rectal organoid morphology analysis (ROMA)—offer functional insights into CFTR activity and drug responsiveness. Compared to existing CFTR functional tests, such as Intestinal Current Measurement (ICM) and Nasal Potential Difference (NPD), these assays are more accessible, highly reproducible, and when needed support personalized medicine approaches. PDIO-based assays could help identify infants at high risk of disease progression, facilitating earlier interventions while minimizing unnecessary follow-ups for those unlikely to develop CF-related symptoms. Although not yet widely implemented, these assays hold promise for refining CFSPID diagnostics and management. Future research should focus on establishing standardized protocols allowing validation of clinical utility.

## 1. Introduction

Cystic fibrosis (CF) is a life-shortening autosomal recessive disease caused by pathological variants (mutations) in the *cystic fibrosis transmembrane conductance regulator* (*CFTR*) gene. It affects multiple organ systems, including the respiratory and gastrointestinal tracts. Early diagnosis of CF is critical, as it improves both quality of life and life expectancy [1]. Consequently, many countries have implemented newborn bloodspot screening (CF-NBS) programs to facilitate early detection and treatment [2,3]. However, CF-NBS also results in cases where the diagnosis remains uncertain, categorized as Cystic Fibrosis Screen Positive Inconclusive Diagnosis/CFTR-related Metabolic Syndrome (CFSPID/CRMS), posing a significant clinical challenge [4,5].

### 1.1. Diagnostic Ambiguity in CFSPID/CRMS

CFSPID/CRMS cases arise when asymptomatic infants exhibit: (1) intermediate sweat chloride concentrations (SCC) and carry at least one CFTR variant of unknown clinical significance (VUS) or with varying clinical consequence (VVCC), (2) when they have a normal SCC but two CFTR variants (of which at least one has unclear phenotypic consequences) [5,6]. The long-term prognosis of these children remains uncertain, and while some eventually convert to a CF diagnosis [7] or CFTR-related disorders (CFTR-RD) [8,9], the majority have no CF (related disease) and will remain asymptomatic [10,11,12]. This diagnostic uncertainty necessitates prolonged follow-up [13], contributing to parental anxiety, overmedicalization and healthcare burden [14].

### 1.2. Current Tools and Limitations in Variant Interpretation

Although genetic testing and CFTR functional assays such as nasal potential difference (NPD) and intestinal current measurement (ICM) have been explored to aid reclassification, their limited accessibility, technical complexity, and non-CFTR dependent biological variability restrict their widespread implementation [15].

In the context of CFSPID/CRMS, extensive genetic testing often reveals VUS and VVCC, which, when found *in trans* with a known CF-causing variant, may result in either CF or CFTR-RD. The clinical interpretation of these variants is complicated by their incomplete penetrance and variable expression across individuals. *CFTR* mutation databases such as the *CFTR2* [16] and *CFTR-France* [17] are valuable resources for assessing the potential clinical relevance of these variants, aiding in more informed decision-making and counseling.

### 1.3. Promise of Patient-Derived Intestinal Organoids (PDIOs)

Recent advances in organoid technology offer a promising alternative to current diagnostic methodologies. Patient-derived intestinal organoids (PDIOs), derived from intestinal stem cells obtained via minimally invasive rectal biopsy, offer a patient-specific model for assessing CFTR function [18]. Several PDIO-based assays are used to quantify CFTR function, including functional assays such as the fluid-transport mediated forskolin-induced swelling (FIS) assay [19,20], and intestinal current measurements on 2D organoid cultures [21]. These assays all rely on CFTR-dependent ion and fluid secretion, yet they are highly complementary as they quantify different ranges of CFTR function [22].

The FIS assay is based on the principle that CFTR activation by forskolin (Fsk) stimulates fluid transport into the organoid lumen, leading to organoid swelling. This response is impaired in people with CF (pwCF) and is commonly used to assess CFTR modulator effects in vitro, enabling sensitive measurements of CFTR function and drug responsiveness while facilitating direct comparisons in PDIOs from pwCF. Clinical studies in pwCF have demonstrated correlations between FIS response and key disease manifestations, including pulmonary function decline, pancreatic phenotypes, and CF-related liver disease. Notably, higher FIS responses are associated with a significant reduced risk of developing a severe clinical CF phenotype [23]. These findings highlight the potential of FIS as a biomarker to stratify for risk of disease severity in pwCF [23].

In organoids from healthy controls or in cases with high residual CFTR function, apical fluid secretion can be observed even without Fsk stimulation, leading to spherical or cystic organoid phenotypes with large fluid-filled lumens. Such forskolin-independent swelling can also be observed when PDIOs with highly responsive CFTR variants are incubated with CFTR modulators. This cystic phenotype can be quantified by the steady-state lumen area (SLA) assay, which measures the lumen surface area as a percentage of total organoid surface area without Fsk stimulation [16]. This SLA assay effectively discriminates between healthy controls and pwCF at the individual level. Among healthy controls, SLA values indicative of the fluid-filled phenotype range from 35% to 70%, with mean values of 51% ± 10 wild-type/wild-type (WT/WT) individuals and 47% ± 11 in WT/F508del carriers. In contrast, pwCF exhibit markedly lower SLA values, typically ranging from 0 to 10%, consistent with a non- to low fluid-filled phenotype. Among CF organoids, SLA discriminates on a group level between PDIOs with class I-III CFTR variants (1.3 ± 1.4) and those with class IV-V CFTR variants (5.0 ± 3.30) (means ± SD) [19]. SLA is therefore particularly useful for evaluating CFTR function in milder disease phenotypes, such as CFSPID/CRMS, CFTR-RD, and other inconclusive diagnosis. Intestinal current measurements (ICM) in PDIO-derived monolayers provide a complementary functional readout by assessing ion transport responses to CFTR-targeting compounds allowing CFTR transport to be quantified as a percentage of normal.

Additionally, new microscopy and automated image analysis techniques are being explored to enhance the throughput, sensitivity, and specificity of these assays. One such approach is the Rectal Organoid Morphology Analysis (ROMA), which is based on the morphological differences between PDIOs from individuals with and without CF [24,25]. ROMA relies on semi-automated image quantification of morphological features in cultured organoids, using image analysis software. The current implementation is operator-guided but software-assisted, meaning standardization and reproducibility depend in part on protocol adherence and user training. Two key parameters are derived from this analysis: the intensity ratio (IR) and the circularity index (CI) of the organoids. IR measures the presence or absence of a central lumen and is higher in CF than in non-CF organoids, while CI quantifies the roundness of the organoids, which is lower in CF than in non-CF organoids. ROMA analysis enables full discrimination of PDIOs from pwCF from those of subjects without CF, providing evidence to support ROMA as a diagnostic test for CF and aiding further classification of inconclusive diagnoses [26].

### 1.4. Advantages of PDIO-Based Assays in Clinical Practice

Compared to other CFTR functional assays such as ICM and NPD, PDIO-based assays offer greater accessibility, as rectal biopsies can be shipped to a reference laboratory for analysis. They are minimally invasive, highly reproducible, and can be performed on cultured cells, allowing for repeated testing without additional biopsies. Moreover, they offer the potential for personalized medicine by enabling drug testing on PDIOs [17], which could help predict responses to CFTR modulators in cases of uncertain clinical progression. Given these advantages, PDIO-based assays may play a crucial role in refining the diagnostic evaluation of CFSPID/CRMS cases. These assays are currently performed in a research setting, with informed consent or assent consistently obtained. However, it is important to acknowledge that rectal biopsy, while generally safe, carries minimal risks and may not be acceptable to all patients or families; thus, alternative CFTR functional tests should also be considered when appropriate.

This article explores the role of PDIO-based assays in the diagnostic work-up of children with a CFSPID/CRMS label, emphasizing their potential to improve early risk stratification and guiding clinical management. Incorporating these assays as additional CFTR functional tests, could help identify infants at low risk of disease progression, reducing unnecessary follow-up while ensuring timely intervention for those at higher risk of conversion to CF or CFTR-RD.

## 2. Materials & Methods

### 2.1. Case Selection and Construction

This article presents four hypothetical CFSPID/CRMS cases, each illustrating a distinct clinical trajectory:Reclassification as Cystic Fibrosis (CF)Reclassification as CFTR-Related Disorder (CFTR-RD)Reclassification as a CF carrier/non-CFUnresolved CFSPID/CRMS label due to inconclusive findings

These cases were designed based on real-life clinical scenarios for which patient-derived intestinal organoid (PDIO)-based assays were requested at either KU Leuven or UMC Utrecht laboratories. While these assays have not yet been routinely applied within the CFSPID context, the clinical cases have been adapted to fit this specific framework to explore the potential utility of these assays. The cases aim to demonstrate the potential diagnostic value of PDIO-based functional assays in CFSPID/CRMS evaluation, addressing the current challenges in predicting disease evolution.

### 2.2. Application of Organoid Assays to Hypothetical Cases

The following assays were described:oRectal Organoid Morphology Analysis (ROMA)oSteady-state lumen area (SLA)oForskolin-Induced Swelling (FIS) assay

These PDIO-based assays were applied to the four cases (reflecting their genotype/phenotype), combining these results with clinical data such as SCC, genetic findings, and clinical symptoms.

## 3. Results

### 3.1. CFSPID Infant Reclassified as Person with CF

#### 3.1.1. Clinical Background

A female infant is labelled as CFSPID/CRMS through CF-NBS. Genetic testing reveals one disease-causing *CFTR* variant (*F508del*) and a VVCC (R117C). Sweat chloride testing at the age of 2 months shows an intermediate level (52 mmol/L).

#### 3.1.2. Follow-Up

The infant is followed closely, and at 2 years of age, she exhibits mild recurrent respiratory infections. Repeat sweat chloride tests show an upward trend, now at 62 mmol/L (CF range).

#### 3.1.3. Role of Organoid-Based Assay

PDIO are derived and tested using:ROMA: PDIOs show CF-like morphology under steady-state culture conditions, consistent with defective CFTR function (Figure 1A,B).SLA: Organoids show a low fluid filled phenotype with a low SLA consistent with a CF organoid phenotype (Figure 1C).FIS assay: Moderate swelling with Fsk alone (without CFTR modulators) indicating some residual function. Significant improvement in organoid swelling with the addition of lumacaftor/ivacaftor and further improvement with elexacator/tezacaftor/ivacaftor (ETI) (Figure 1D).

#### 3.1.4. Outcome & Impact

Based on the combination of worsening clinical symptoms, sweat chloride levels, and impaired CFTR function in organoids, the patient is reclassified as having CF. While the point-of-care utility of patient-derived intestinal organoid (PDIO) assays is well illustrated in this case, it is important to acknowledge that a clinical diagnosis of CF can be made based on the triad of suggestive symptoms, sweat chloride testing, and genetic findings—even in the absence of functional CFTR assays. However, the inclusion of PDIO testing in the diagnostic work-up can provide additional reassurance regarding classification decisions. In this case, earlier use of PDIO testing could potentially have supported earlier reclassification and initiation of CF-specific treatments, which may contribute to improved long-term outcomes.

### 3.2. CFSPID Infant Reclassified as CFTR-RD

#### 3.2.1. Clinical Background

A male infant is labelled as CFSPID/CRMS through CF-NBS. Genetic testing reveals one disease-causing *CFTR* variant (*F508del*) and one CFTR-RD causing variant (*N1303I*). SCC at the age of 2 months is in the intermediate range (41 mmol/L).

#### 3.2.2. Follow-Up Challenges

At a later age, he develops failure to thrive and, at the age of 14-months, presents with a severe metabolic alkalosis consistent with pseudo-Bartter syndrome. No other overt CF symptoms are present. A repeat sweat test remains in the intermediate range (35 mmol/L).

#### 3.2.3. Role of Organoid-Based Assay

PDIO are derived and tested using:ROMA: PDIOs exhibit CF-like morphology, indicative of defective CFTR function (Figure 1A,B).SLA: Organoids show a low fluid filled morphology with a low SLA, also indicative of defective CFTR function (Figure 1C).FIS assay: Stimulation of CFTR with Fsk alone leads to significant organoid swelling, indicating high residual function. Swelling increases slightly by the CFTR modulators tezacaftor/ivacaftor but does not further increase with the addition of elexacaftor (Figure 1D).

#### 3.2.4. Outcome & Impact

Given the child’s pseudo-Bartter syndrome, stable SCC, and lower than normal CFTR function on organoid assays, he has been reclassified as having CFTR-RD. Following appropriate treatment and monitoring, the child recovers fully. The child remains free of symptoms during the first 3 years of follow-up on a salt-rich diet.

A high FIS level indicates a significantly reduced risk of developing a severe clinical CF phenotype, but follow-up at a CF center is still recommended due to the potential for symptom development. If organoid testing had been conducted earlier, detecting partial CFTR dysfunction would have supported continued follow-up, rather than discharge. This case highlights the importance of monitoring children at risk of developing CF or CFTR-related disorder (CFTR-RD) at a specialized CF center, where clinical expertise and early recognition of evolving symptoms enable timely diagnosis and appropriate management.

### 3.3. CFSPID Infant Reclassified as CF Carrier/Non-CF

#### 3.3.1. Clinical Background:

A female infant is labeled CFSPID/CRMS after a positive CF-NBS result along with a SCC of 38 mmol/L. Genetic testing reveals a VUS in addition to the previously disease-causing *CFTR* variant (*F508del*).

#### 3.3.2. Follow-Up Challenges:

SCC normalizes 2 years of age.

#### 3.3.3. Role of Organoid-Based Assay

PDIO are derived and tested using:ROMA: Morphology matches non-CF controls (Figure 1A,B).SLA: Organoids show a fluid-filled phenotype with SLA consistent with wild-type status (Figure 1C).FIS assay: Little swelling is measured during the assay as organoids are already swollen without addition of Fsk or CFTR-modulators. Swelling increases slightly by adding Fsk, CFTR modulators perform similar or worse (Figure 1D).

#### 3.3.4. Outcome & Impact

This child has had a positive IRT and a borderline sweat chloride concentration (SCC). While PDIO assays confirm normal CFTR function, strongly supporting the conclusion that this child is a CF carrier, some residual uncertainty remains. As such, while the child is discharged from CF specialist care for now with follow-up managed in primary care, clinicians should remain vigilant. If any symptoms emerge or diagnostic uncertainty arises, referral back to a CF center for repeat sweat testing is recommended.

### 3.4. Case 4: CFSPID Infant Who Remains Inconclusive

#### 3.4.1. Clinical Background

A male infant is categorized as CFSPID/CRMS after a positive CF-NBS with an intermediate SCC (50 mmol/L). Genetic testing revealed one CF-causing variant (*F508del*) and one VVCC (*R117H-7T*).

#### 3.4.2. Follow-Up Challenges

The child remains asymptomatic but has fluctuating SCC ranging from 45 to 55 mmol/L over time. Clinicians are unsure whether he will develop CF or CFTR-RD.

#### 3.4.3. Role of Organoid-Based Assay

PDIO are derived and tested using:ROMA: Morphology falls into an intermediate range (region of uncertainty), making classification difficult (Figure 1A,B).SLA: SLA falls into intermediate range, making classification difficult (Figure 1C).FIS assay: Stimulation of CFTR with Fsk alone causes organoid swelling (>3000 FIS units) indicating high residual function and therefore low risk of developing a severe clinical CF phenotype. Swelling decreases with all CFTR modulators (Figure 1D).

#### 3.4.4. Outcome & Impact

The unclear PDIO assay results mirror the borderline clinical findings, meaning the child remains classified as CFSPID/CRMS. Continued follow-up in accordance with CFSPID/CRMS guidelines is recommended.

## 4. Discussion

This educational article explored the role of PDIO-based assays in the diagnostic work-up of children with a CFSPID/CRMS label. The four hypothetical cases illustrate the promising role of PDIO assays in early risk classification. In the first case, an infant with intermediate sweat chloride and two *CFTR* variants exhibited minimal CFTR function in PDIO assays and was later reclassified as CF. Early identification of impaired CFTR function could have facilitated closer monitoring and earlier intervention. In the second case, an infant with intermediate SCC and a VUS showed partial CFTR function and was later reclassified as CFTR-RD. PDIO testing could have guided personalized follow-up. The third case involved an infant with an ambiguous genetic profile and near-normal CFTR function who remained symptom-free and was ultimately classified as a CF carrier, suggesting that PDIO assays could help discontinue unnecessary follow-up. The final case remained inconclusive despite intermediate CFTR function in PDIO, highlighting both the limitations of PDIO-based assays and their potential role in informing long-term monitoring strategies when diagnostic uncertainty persists. However, based on the FIS levels measured in PDIOs, the risk of this patient developing severe CF disease is considered to be very low [23].

CFSPID/CRMS represents a diagnostic challenge due to its heterogeneous outcomes, ranging from eventual CF diagnosis to individuals remaining asymptomatic and well without clinical consequences [9,10,11]. Currently, follow-up relies primarily on sweat chloride testing and genetic analysis [12], which may not always provide definitive answers. This paper highlights the potential role of PDIO-based assays in enhancing risk stratification for CFSPID/CRMS infants by directly assessing CFTR function, helping differentiate those at high and low risk for disease progression, and ultimately improving clinical decision-making. The cases discussed demonstrate the clinical implications of integrating PDIO-based assays into CFSPID/CRMS diagnostics, where for high-risk infants, these assays could identify impaired CFTR function, supporting closer monitoring and earlier treatment with CFTR modulators. Conversely, for infants who do not develop CF, functional testing could enable earlier identification and safe discharge from follow-up programs, thereby reducing prolonged diagnostic uncertainty and its psychological and healthcare impacts. In this context, biomarkers of CFTR function, such as those derived from PDIO-based assays, can add an important layer of evidence to support clinical decisions regarding the intensity and duration of follow-up in CFSPID/CRMS cases.

Although PDIO assays, such as ROMA, SLA and FIS are not yet routinely used in CFSPID/CRMS diagnostics, they offer significant potential for improving early risk stratification, reducing unnecessary follow-up, and refining long-term management strategies. However, their clinical utility still requires validation through further studies to confirm their accuracy and predictive value. Additionally, standardized protocols must be developed to ensure consistent application across clinical settings. Future research is essential to establish how these assays can be effectively incorporated into routine practice, potentially enabling a more personalized approach to CF care, and ensuring each child receives the most appropriate surveillance and intervention based on their individual risk profile.

## Figures and Tables

**Figure 1 IJNS-11-00052-f001:**
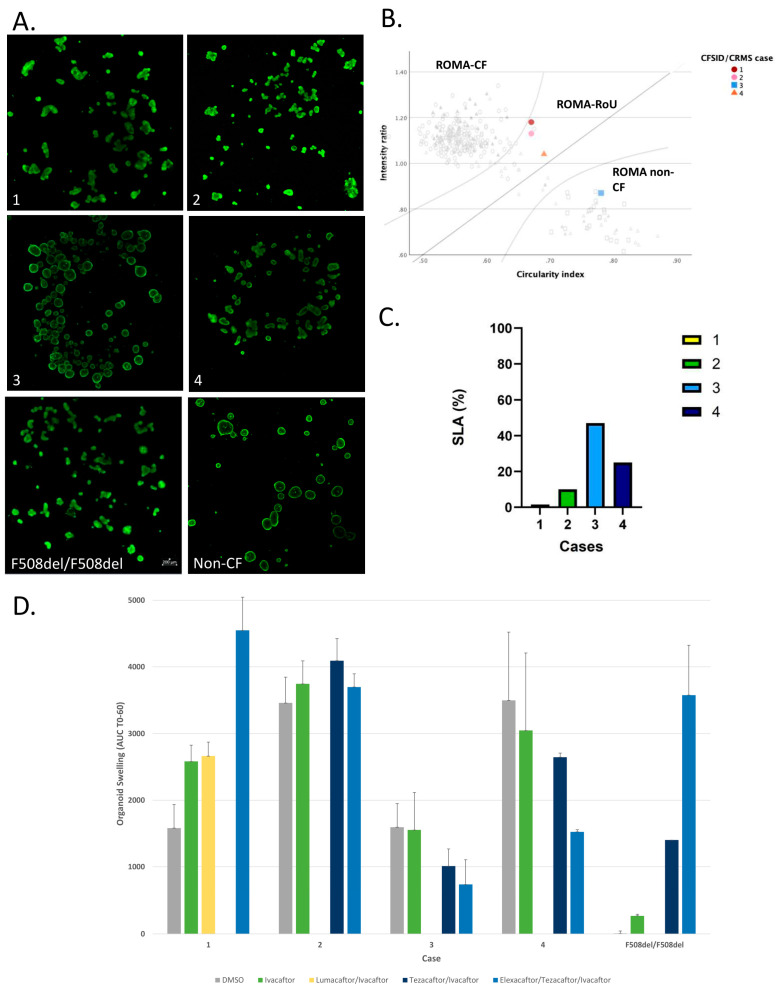
PDIO-assays for CFSPID/CRMS cases. (**A**) Images of PDIO. Cases 1 and 2 exhibit irregular shaped organoids with no visible lumen in the majority of PDIOs. Case 3 exhibits circular and swollen organoids. Case 4 exhibits a mixed morphology. PDIOs from F508del homozygous person with CF and from a healthy control are shown for reference. (**B**) ROMA analysis result. Cases 1 and 2 are categorized as ROMA-CF while case 3 is categorized as ROMA non-CF. Case 4 falls in the ROMA Region of Uncertainty (RoU). Grayscale image reproduced from Cuyx, S.; Santo Ramalho, A.; Fieuws, S.; Corthout, N.; Proesmans, M.; Boon, M.; Arnauts, K.; Carlon, M.S.; Munck, S.; Dupont, L.; et al. Rectal organoid morphology analysis (ROMA) as a novel physiological assay for diagnostic classification in cystic fibrosis. *Thorax 79*, 834–841, Copyright 2025 with permission from BMJ Publishing Group Ltd. [26] is shown in light grey for reference. (**C**) SLA analysis result. Case 1 displays SLA values consistent with a CF organoid phenotype. Case 2 exhibits reduced SLA, indicating impaired CFTR function with residual function. Case 3 shows SLA consistent with wild-type CFTR function. Case 4 demonstrates intermediate SLA values, falling between CF and non-CF ranges. (**D**) FIS assay results. Responses to Fsk 0.8 µM and CFTR modulators are shown per case. Lumacaftor/Ivacaftor was only tested in case 1, Tezacaftor/Ivacaftor was tested in the other cases. FIS assay results from a *F508del* homozygous patient are shown for reference. Figure 1A,B,D for case 2 adapted from Rodriguez Mier et al. (2025) [27]. Reprinted from *Journal of Cystic Fibrosis*, Vol. 24, No. 2, Rodriguez Mier, N., Antoons, V., Cuyx, S., Santo Ramalho, A., Boon, M., Proesmans, M., … & Vermeulen, F., Pseudo-Bartter syndrome: A CFTR-related disorder?, pp. 401–403, Copyright (2025), with permission from Elsevier.

## Data Availability

The original contributions presented in this study are included in the article. Further inquiries can be directed to the corresponding author.

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
