# Peer review of "The Promising Role of Intestinal Organoids in the Diagnostic Work-Up of Cystic Fibrosis Screen Positive Inconclusive Diagnosis/CFTR-Related Metabolic Syndrome (CFSPID/CRMS)"

_2409-515X, 2025, doi:10.3390/ijns11030052_

Round 1
Reviewer 1 Report (Previous Reviewer 1)
Comments and Suggestions for Authors
Thank you for the responses to reviewer comments – the adjustments have made for a stronger piece of writing. I also believe that this is better aligned with a commentary rather than review article format.
Minor points:
- Line 329: the term ‘resolution’ is probably slightly mis-placed here, since there are no symptoms or tests that ‘resolve’ (likely the only positive is an IRT and that is not repeated). The outcomes range from eventual CF diagnosis to continuing to remain well.
- The problem with Error messages appears to have arisen again from page 3 onwards. Presumably a problem with referencing software.
Author Response
Thank you for your comments.
Comment 1: Line 329: the term ‘resolution’ is probably slightly mis-placed here, since there are no symptoms or tests that ‘resolve’ (likely the only positive is an IRT and that is not repeated). The outcomes range from eventual CF diagnosis to continuing to remain well.
Reply: That is true. We've adapted to: "CFSPID/CRMS represents a diagnostic challenge due to its heterogeneous outcomes, ranging from eventual CF diagnosis to individuals remaining asymptomatic and well without clinical consequences".
Comment 2: The problem with Error messages appears to have arisen again from page 3 onwards. Presumably a problem with referencing software
Reply: Probably indeed. This issue was fixed.
Reviewer 2 Report (New Reviewer)
Comments and Suggestions for Authors
-I think I saw the revised version with the suggestions of two referees. This version is very easy to read, very clear, and well written.
-"Error! Bookmark not defined.." is present in some places, please check
- A small suggestion, the examination results of the patients and their results can also be given in a small table.
Author Response
Thank you for your comments.
Comment 1: I think I saw the revised version with the suggestions of two referees. This version is very easy to read, very clear, and well written.
Reply: Yes, indeed you did. Thank you very much.
Comment 2: "Error! Bookmark not defined.." is present in some places, please check.
Reply: On pages 3,5 and 9. This issue was fixed.
Comment 3: A small suggestion, the examination results of the patients and their results can also be given in a small table.
Reply: Thank you for your suggestion. However, we feel that presenting the results in the current narrative format allows for clearer context and avoids overcomplicating the section, given the limited number of cases. I hope that approach works for this purpose. Furthermore, everything is visually summarised in Figure 1.
This manuscript is a resubmission of an earlier submission. The following is a list of the peer review reports and author responses from that submission.
Round 1
Reviewer 1 Report
Comments and Suggestions for Authors
General comments
Thank you for your article. This is a well written piece discussing an important area for future work in CFTR diagnostics, and will be of interest to people working in this field. There are some areas in which I would like to see the certainty around the language adjusted, acknowledging where uncertainty still exists. Whether the style of the review (case examples rather than true literature review) meets the criteria of the journal is an editorial decision.
Major comments
- Where the concept of CFSPID/CRMS is first discussed, it needs to be emphasised that alongside the genetic/ SCC criteria for this designation, the overarching designation is predicated on the fact that the child is asymptomatic. Please mention this, perhaps on line 49?
- I think I can understand the point that you are making with the sentence commencing on line 93, but it is long and challenging to follow. I suggest breaking up the sentence to make it more clear that there are 2 groups within the healthy controls – there is an overall larger proportion of the cystic phenotype, and that this is slightly different between WT/WT and carriers? I think the term fluid-filled/cystic phenotype is confusing as I believe you are describing the same thing – just stick with one or the other?
- The introduction would benefit from some subheadings to help demarcate the specific points or methods being used
- The authors describe the ROMA methodology, though this would benefit from a little more detail about how this methodology works – is this a machine learning AI approach or very much operator driven? In addition to stating that ROMA is good at differentiating CF vs non-CF, there should be a mention in this section (Line 104-114) of whether any studies describe its utility in cases of CFSPID or CFTR-related disorders.
- ‘Reclassification as non-CF’ should probably be described as reclassification as a CF-carrier? These terms are used slightly interchangeably during that case, so would be good to be consistent.
- Some of the organoids in the images of Figure 1A are difficult to distinguish. Is there a way of making these more clear, perhaps in the green form sometimes shown at meetings? Figure 1D also has slightly blurry text.
- The point made about PDIO utility in 3.1.4 is valid, though it is worth acknowledging (for completeness and for those who do not have access to PDIO or other CFTR functional tests) that the combination of genetics, symptoms and SCC alone are enough for a clinical diagnosis of CF to be made. In this hypothetical case, PDIO is reassuring that this re-classification is correct, and could potentially have helped reclassify earlier without waiting for a rise in SCC.
- For case 2 (3.2 in text), I think the wording/ messaging needs to be careful here. The genotype (F508del/ D1152H) is well described in association with low/ borderline SCC, irrespective of clinical symptoms and ultimate diagnosis – this is thought to be related to abnormal bicarbonate rather that chloride ion secretion. It is not completely clear from the text what makes the PDIO different to the first case, and therefore why could this not be CF even in the presence of a SCC <60mmol/L (such cases are described in the literature, even with CFTR functional testing such as NPD) – presumably the reduced response to ETI? Are we sure that D1152H responds in a predictable way in the PDIO assays commensurate with clinical phenotype, given the slightly ‘special’ nature of SCC? Additionally, the comment about ‘High FIS level indicating reduced risk of a severe CF phenotype’ – it is widely known that VVCC’s in pancreatic sufficient CF have an overall milder phenotype. I also dispute that earlier testing would have changed the clinical outcome here – CFSPID or CFTR-RD in most centres will both be seen infrequently.
- 3.2 needs elaboration. In previous cases there has been some description on clinical progress – well/ not well etc – and this should be no different.
- Regarding case 3: How sure are we that there is no development of disease in later life for this case (F508del/ L101S)? Are there long-term follow-up data to support this, or data from adult bronchiectasis cohorts? This hypothetical child has had a positive IRT and a borderline SCC – are we sure that they are a carrier? If not, perhaps this article should advocate for a ‘CFSPID light’ approach whereby a SCC is rechecked age 6 and ? as a teenager? We know that SCC varies quite significantly within each patient, so who is to say that the SCC would not be 38mmol/L later on and, if so, would that affect the choice to discharge? I would recommend that the authors are less definitive in areas where we do not have complete knowledge, and can do this whilst also advocating for the use of PDIO.
- In 3.3.4 – please explain the relevance of swelling decreasing with all CFTR modulators.
- This article clearly advocates for the widespread use of PDIO testing in CFSPID/CRMS, though there is no discussion of the potential risks of doing so – they are minimal, but rectal biopsy is not without its risks and will be unacceptable for some parents/ children. This should be acknowledged in the context of alternative CFTR functional tests.
- It is not clear to this reviewer whether the data presented in Figure 1 comes from patients with the genotype/ phenotype described in the cases, or are completely unrelated examples of PDIO data. Please clarify this in the text around line 140.
Minor points
- Sentence on Line 21/22 doesn’t make grammatical sense
- Line 56 – the point made is valid, particularly prolonged follow-up, though in theory the consensus guidelines do not recommend “intensive” follow-up but rather infrequent reviews
- Line 65 – should read CFTR mutation databases
- The sentence beginning on line 84 (finishing on line 85) needs to be referenced
- There are several error messages throughout the text (inc line 52, 93, 98, 121) – please correct. I assume that they are problematic references.
- Section 2.2 has a repeated sentence
- Should RuO in Figure 1 legend read RoU?
Reviewer 2 Report
Comments and Suggestions for Authors
This is a well-written manuscript that attempts to provide a clinical application for intestinal organoids. The authors are based at one of the leading organoid centers in Europe and have led the field in terms of developing organoid techniques and translational interpretation. They are applying their expertise to hypothetical clinical cases of CFSPID in which the diagnosis of cystic fibrosis, clinical course, and need for therapeutic monitoring. The appeal of their research is quite evident.
The major strength of the manuscript is the authors' expertise in interpretation of organoid swelling, multiple assays of CFTR function, and the clinical question of CFSPID of uncertain disease trajectory.
The weakness of the manuscript is that to my understanding it presents 4 hypothetical cases of CFSPID with hypothetical organoid responses. It is not clear to me if the authors are presenting new data or just further interpreting potential clinical contexts where the panel of swelling assays (ROMA, SLA, and FIS) might be helpful. A second consideration is the potential overlap of this publication with the Cuyx 2024 Thorax publication. Although this is more focused on CFSPID and the other is on CF vs non-CF the overlap is clear. The appeal of this manuscript would be enhanced if they were to show new data of pts in the 4 clinical contexts rather than just hypothetical considerations.
The question then is how to improve the manuscript to increase enthusiasm for publication. The biggest improvement would be the inclusion of actual organoid data from CFSPID subjects that is already available within the authors repository of specimens. A second option would be to be more overt that these are just hypothetical considerations rather than presentation of actual data. Currently, the article is listed as a "review," but it might be better listed as a short letter or commentary on an additional application of their previous findings. Perhaps that would be better stated in the title, abstract, and data.
A small comment is that for some reason "Error! Bookmark not defined.." is present in multiple places in the manuscript. I suspect this is regarding citations, but should be addressed.
Overall, the utilization of an organoid assay panel to categorize CFSPID patients is appealing. However, the manuscript would be more impactful if connected to actual patient data and outcomes.
